# Smoothed Geometry for Robust Attribution

**Zifan Wang**
Electrical and Computer Engineering
Carnegie Mellon University
zifan@cmu.edu

**Haofan Wang**
Electrical and Computer Engineering
Carnegie Mellon University

**Shakul Ramkumar**
Information Networking Institute
Carnegie Mellon University

**Matt Fredrikson**
School of Computer Science
Carnegie Mellon University

**Piotr Mardziel**
Electrical and Computer Engineering
Carnegie Mellon University

**Anupam Datta**
Electrical and Computer Engineering
Carnegie Mellon University

## Abstract

Feature attributions are a popular tool for explaining the behavior of Deep Neural Networks (DNNs), but have recently been shown to be vulnerable to attacks that produce divergent explanations for nearby inputs. This lack of robustness is especially problematic in high-stakes applications where adversarially-manipulated explanations could impair safety and trustworthiness. Building on a geometric understanding of these attacks presented in recent work, we identify Lipschitz continuity conditions on models' gradients that lead to robust gradient-based attributions, and observe that the *smoothness* of the model's decision surface is related to the transferability of attacks across multiple attribution methods. To mitigate these attacks in practice, we propose an inexpensive regularization method that promotes these conditions in DNNs, as well as a stochastic smoothing technique that does not require re-training. Our experiments on a range of image models demonstrate that both of these mitigations consistently improve attribution robustness, and confirm the role that smooth geometry plays in these attacks on real, large-scale models.

## 1 Introduction

Attribution methods map each input feature of a model to a numeric score that quantifies its relative importance towards the model's output. At inference time, an analyst can view the attribution map alongside its corresponding input to interpret the data attributes that are most relevant to a given prediction. In recent years, this has become a popular way of explaining the behavior of Deep Neural Networks (DNNs), particularly in domains such as medical imaging [5] and other safety-critical tasks [23] where the opacity of DNNs might otherwise prevent their adoption.

Recent work has shown that attribution methods are vulnerable to adversarial perturbations [12, 14, 16, 19, 47], showing that it is often possible to find a small-norm set of feature changes that yield attribution maps with adversarially-chosen qualities while leaving the model's output behavior intact. For example, an attacker might introduce visually-imperceptible changes that cause the mapping generated for a medical image classifier to focus attention on an irrelevant region. Such attacks have troubling implications for the continued adoption of attribution methods for explainability in high-stakes settings.

**Contributions**. In this paper, we characterize the vulnerability of attribution methods in terms of the geometry of the targeted model's decision surface. Restricting our attention to attribution methods that primarily use information from the model's gradients [38, 41, 45], we formalize attribution robustness as a local Lipschitz condition on the mapping, and show that certain smoothness criteria of the model ensure robust attributions (Sec. 3, Theorems 1 and 2). Importantly, our analysis suggests that attacks are less likely to transfer across attribution methods when the model's decision surface is smooth (Sec. 3.2), and our experimental results confirm this on real data (Sec. 5.3). While this phenomenon is widely-known for adversarial examples [46], to our knowledge this is the first systematic demonstration of it for attribution attacks.

As typical DNNs are unlikely to satisfy the criteria we present, we propose *Smooth Surface Regularization* (SSR) to impart models with robust gradient-based attributions (Sec. 4, Def. 7). Unlike prior regularization techniques that aim to mitigate attribution attacks [8], our approach does not require solving an expensive second-order inner objective during training, and our experiments show that it effectively promotes robust attribution without a significant reduction in model accuracy (Sec. 5.2). Finally, we propose a stochastic post-processing method as an alternative to SSR (Sec. 4), and validate its effectiveness experimentally on models of varying size and complexity, including pre-trained ImageNet models (Sec. 5.1).

Taken together, our results demonstrate the central role that model geometry plays in attribution attacks, and that a variety of techniques that promote smooth geometry can effectively mitigate the problem on large-scale, state-of-the-art models. Proofs for all theorems and propositions in this paper are included in Appendix A and the implementation is available on: `https://github.com/zifanw/smoothed_geometry`

## 2 Background

We begin with notation, and proceed to introduce the attribution methods considered throughout the paper, as well as and the attacks that target them. Let $\arg\max_c f_c(\mathbf{x}) = y$ be a DNN that outputs a predicted class $y$ for an input $\mathbf{x} \in \mathbb{R}^d$. Unless stated otherwise, we assume that $f$ is a feed-forward network with rectified-linear (ReLU) activations.

**Attribution methods.** An attribution $\mathbf{z} = g(\mathbf{x}, f)$, indicates the importance of features $\mathbf{x}$ towards a *quantity of interest* [22] $f$, which for our purposes will be the pre- or post-softmax score of the model's predicted class $y$ at $\mathbf{x}$. When $f$ is clear from the context, we write $g(\mathbf{x})$. We also denote $\nabla_{\mathbf{x}} f(\mathbf{x})$ as the gradient of $f$ w.r.t the input. Throughout the paper we focus on the following gradient-based attribution methods.

**Definition 1** (Saliency Map (SM) [38]). *Given a model $f(\mathbf{x})$, the Saliency Map for an input $\mathbf{x}$ is defined as $g(\mathbf{x}) = \nabla_{\mathbf{x}} f(\mathbf{x})$.*

**Definition 2** (Integrated Gradients (IG) [45]). *Given a model $f(\mathbf{x})$, a user-defined baseline input $\mathbf{x}_b$, the Integrated Gradient is the path integral defined as $g(\mathbf{x}) = (\mathbf{x} - \mathbf{x}_b) \circ \int_0^1 \nabla_{\mathbf{r}} f(\mathbf{r}(t)) dt$ where $\mathbf{r}(t) = \mathbf{x}_b + (\mathbf{x} - \mathbf{x}_b)t$ and $\circ$ is Hadamard product.*

**Definition 3** (Smooth Gradient (SG) [41]). *Given $f(\mathbf{x})$ and a user-defined variance $\sigma$, the Smooth Gradient is defined as $g(\mathbf{x}) = \mathbb{E}_{\mathbf{z} \sim \mathcal{N}(\mathbf{x}, \sigma^2 \mathbf{I})} \nabla_{\mathbf{z}} f(\mathbf{z})$.*

We focus our attention on the above three methods as they are widely available, e.g., as part of Pytorch's Captum [20] API, and they work across a broad range of architectures—a property that many other methods (e.g., DeepLIFT [36], LRP [6], and various CAM-based methods [30, 35, 49, 54]) do not satisfy. We exclude Guided Backpropogation [44] as it may not be sensitive to the model [1], as well as perturbation methods [33, 34, 40, 52], as they have not been the focus of prior attacks.

**Attacks.** Similar to *adversarial examples* [15], recent work demonstrates that gradient-based attribution maps are also vulnerable to small distortions [19]. We refer to an attack that tries to modify the original attribution map with properties chosen by the attacker as *attribution attack*.

**Definition 4** (Attribution attack). *A an attribution method $g$ for model $f$ is vulnerable to attack at input $\mathbf{x}$ if there exists a perturbation $\boldsymbol{\epsilon}$, $||\boldsymbol{\epsilon}|| \leq \delta_p$, where $g(\mathbf{x}, f)$ and $g(\mathbf{x} + \boldsymbol{\epsilon})$ are dissimilar but the model's prediction remains unchanged. An attacker generates $\boldsymbol{\epsilon}$ by solving Equation 1:*

$$\min_{||\boldsymbol{\epsilon}||_p \leq \delta} \mathcal{L}_g(\mathbf{x}, \mathbf{x} + \boldsymbol{\epsilon}) \quad s.t. \arg\max_c f_c(\mathbf{x}) = \arg\max_c f_c(\mathbf{x} + \boldsymbol{\epsilon}) \tag{1}$$

*where $\mathcal{L}_g$ is an attacker-defined loss measuring the similarity between $g(\mathbf{x})$ and $g(\mathbf{x} + \boldsymbol{\epsilon})$.*

To perform an attribution attack for ReLU networks, a common technique is to replace ReLU activations with an approximation whose second derivatives are non-zero, $\mathrm{S}(\mathbf{x}) \overset{\text{def}}{=} \beta^{-1}[1 + \exp(\beta\mathbf{x})]$. Ghorbani et al. [14] propose the *top-k*, *mass-center*, and *targeted* attacks, each characterized by different $\mathcal{L}_g$. As an improvement to the targeted attack, the *manipulate* attack [12] adds a constraint to $\mathcal{L}_g$ that promotes similarity of the model's output behavior between the original and perturbed inputs, beyond prediction.

## 3 Characterization of Robustness

In this section, we first characterize the robustness of attribution methods to these attacks geometrically, primarily in terms of Lipschitz continuity. In Section 3.1, we build on this characterization to show why, and under what conditions, the Smooth Gradient is more robust than a Saliency Map. Finally, Section 3.2 discusses the transferability of attribution attacks, leveraging this geometric understanding of vulnerability to shed light on the conditions that make it more likely to succeed.

**Robust Attribution.** The attacks described by Definition 4 can all be addressed by ensuring that the attribution map remains stable around the input $\mathbf{x}$, which motivates the use of Lipschitz continuity (Definition 5) to measure attribution robustness [2, 3] (Definition 6). Unless otherwise noted, we assume that $p = 2$ when referring to Definitions 5 and 6.

**Definition 5** (Lipschitz Continuity). *A general function $h : \mathbb{R}^{d_1} \to \mathbb{R}^{d_2}$ is $(L, \delta_p)$-locally Lipschitz continuous if $\forall \mathbf{x}' \in B(\mathbf{x}, \delta_p), ||h(\mathbf{x}) - h(\mathbf{x}')||_p \leq L||\mathbf{x} - \mathbf{x}'||_p$. Similarly, $h$ is $L$-globally Lipschitz continuous if $\forall \mathbf{x}' \in \mathbb{R}^{d_1}, ||h(\mathbf{x}) - h(\mathbf{x}')||_p \leq L||\mathbf{x} - \mathbf{x}'||_p$*

**Definition 6** (Attribution Robustness). *An attribution method $g(\mathbf{x})$ is $(\lambda, \delta_p)$-locally robust if $g(\mathbf{x})$ is $(\lambda, \delta_p)$-locally Lipschitz continuous, and $\lambda$-globally robust if $g(\mathbf{x})$ is $\lambda$-globally Lipschitz continuous.*

Viewing the geometry of a model's boundaries in its input space provides insight into why various attribution methods may not be robust. We begin by analyzing the robustness of Saliency Maps in this way. Recalling Def. 1, a Saliency Map represents the steepest direction of output increase at $\mathbf{x}$. However, as ReLU networks are typically very non-linear and the decision surface is often concave in the input space, this direction is usually only consistent in an exceedingly small neighborhood around $\mathbf{x}$. This implies weak robustness, as the $\delta_p$ needed to satisfy Definition 6 will be too small for most practical scenarios [19]. Theorem 1 formalizes this, bounding the robustness of the Saliency Map in terms of the local Lipschitz continuity of the model at $\mathbf{x}$, and Example 1 provides further intuition with a low-dimensional illustration.

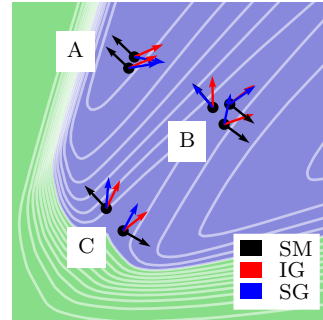

Figure 1: Attributions normalized to unit length in two-dimensions. Score surface is represented by contours. Green and purple areas are two predictions.

**Theorem 1.** *Given a model $f(\mathbf{x})$ is $(L, \delta_2)$-locally Lipschitz continuous in a ball $B(\mathbf{x}, \delta_2)$, the Saliency Map is $(\lambda, \delta_2)$-locally robustness where the upper-bound of $\lambda$ is $O(L)$.*

**Example 1.** *Fig. 1 shows the decision surface of an example two-dimensional ReLU network as a contour map, and the network's binary prediction boundary with green and purple regions. The vectors represent the Saliency Map (black), Integrated Gradient (red), and Smooth Gradient (blue) for inputs in different neighborhoods ($A$, $B$ and $C$). The attributions are normalized to unit length, so the difference in the direction is proportional to the corresponding $\ell_2$ distance of $g(\mathbf{x}, f)$ across the three methods [9]. Observe that for points in the same neighborhood, the local geometry determines whether similar inputs receive similar attributions; therefore, attribution maps in $A$ are more robust than those in $B$ and $C$, which happen to sit on two sides of the ridge.*

### 3.1 Robustness of Stochastic Smoothing

Local geometry determines the degree of robustness of the Saliency Map for an input. To increase robustness, an intuitive solution is to smooth this geometry, therefore increasing local-Lipschitz

continuity. Dombrowiski et al. [12] firstly prove that Softplus network has more robust Saliency Map than ReLU networks. Given a one-hidden-layer ReLU network $f_r$ and its counterpart $f_s$ obtained by replacing ReLU with Softplus, they further observe that with a reasonable choice of $\beta$ for the Softplus activation, the Saliency Map on $f_s$ is a good approximation to the Smooth Gradient on $f_r$. By approximating Smooth Gradient with Softplys, they managed to explain why Smooth Gradient may be more robust for a simple model. In this section, we generalize this result, and show that SmoothGrad can be calibrated to satisfy Def. 6 for arbitrary networks. We begin by introducing Prop 1, which establishes that applying Smooth Gradient to $f(\mathbf{x})$ is equivalent to computing the Saliency Map of a different model, obtained by convolving $f$ with isotropic Gaussian noise.

**Proposition 1.** *Given a model $f(\mathbf{x})$ and a user-defined noise level $\sigma$, the following equation holds for Smooth Gradient: $g(\mathbf{x}) = \mathbb{E}_{\mathbf{z} \sim \mathcal{N}(\mathbf{x}, \sigma^2 \mathbf{I})} \nabla_{\mathbf{z}} f(\mathbf{z}) = \nabla_{\mathbf{x}}[(f * q)(\mathbf{x})]$ where $q(\mathbf{x}) \sim \mathcal{N}(\mathbf{0}, \sigma^2 \mathbf{I})$ and $*$ denotes the convolution operation [7].*

The convolution term reduces to Gaussian Blur, a widely-used technique in image denoising, if the input has only 2 dimensions. Convolution with Gaussian noise smooths the local geometry, and produces a more continuous gradient. This property is formalized in Theorem 2

**Theorem 2.** *Given a model $f(\mathbf{x})$ where $\sup_{\mathbf{x} \in \mathbb{R}^d} |f(\mathbf{x})| = F < \infty$, Smooth Gradient with standard deviation $\sigma$ is $\lambda$-globally robust where $\lambda \leq 2F/\sigma^2$.*

Theorem 2 shows that the global robustness of Smooth Gradient is $O(1/\sigma^2)$ where a higher noise level leads to a more robust Smooth Gradient. On the other hand, lower supremum of the absolute value of output scores will also deliever a more robust Smooth Gradient. To explore the local robustness of Smooth Gradient, we associate Theorem 1 to locate the condition when SmoothGrad is more robust than Saliency Map.

**Proposition 2.** *Let $f$ be a model where $\sup_{\mathbf{x} \in \mathbb{R}^d} |f(\mathbf{x})| = F < \infty$ and $f$ is also $(L, \delta_2)$-locally Lipschitz continuous in the ball $B(\mathbf{x}, \delta_2)$. With a proper chosen standard deviation $\sigma > \sqrt{\delta_2 F/L}$, the upper-bound of the local robustness of Smooth Gradient is always smaller than the upper-bound of the local robustness of Saliency Map.*

**Remark**. Upper-bounds of the robustness coefficients describe the worst-case dissimilarity between attributions for nearby points. The least reasonable noise level is proportional to $\sqrt{1/L}$. When we fix the size of the ball, $\delta_2$, Saliency Map in areas where the model has lower Lipschitz continuity constant is already very robust according to Theorem 1. Therefore, to significantly outperform the robustness of Saliency Map in a scenario like this, we need a higher noise level $\sigma$. For a fixed Lipschitz constant $L$, achieving local robustness across a larger local neighborhood requires proportionally larger $\sigma$.

### 3.2 Transferability of Local Perturbations

Viewing Prop. 2 from an adversarial view, it is possible that an adversary happens to find a certain neighbor whose local geometry is totally different from the input so that the chosen noise level is not large enough to produce semantically similar attribution maps. Similar idea can also be applied to Integrated Gradient. Region B in the Example 1 shows that how tiny local change can affect the gradient information between the baseline and the input: when two nearby points are located on each side of the ridge, the linear path from the baseline (left bottom corner) towards the input can be very different. The possibility of a transferable adversarial noise is critical since attacking Saliency Map requires much less computation budget than attacking Smooth Gradient and Integrated Gradient. We demonstrate the transfer attack of attribution maps and discuss the resistance of different methods against transferred adversarial noise in the Experiment III of Sec. 5.

## 4 Towards Robust Attribution

In this section, we propose a remedy to improve the robustness of gradient-based attributions by using Smooth Surface Regularization during the training. Alternatively, without retraining the model, we discuss Uniform Gradient, another stochastic smoothing for the local geometry, towards robust interpretation. Gradient-based attributions relates deeply to Saliency Map by definitions. Instead of directly improving the robustness of Smooth Gradient or Integrated Gradient, which are computationally intensive, we should simply consider how to improve the robustness of Saliency

Map, the continuity of input gradient. Theorem 3 builds the connection between the continuity of gradients for a general function with the input Hessian.

**Theorem 3.** *Given a twice-differentiable function* $f : \mathbb{R}^{d_1} \to \mathbb{R}^{d_2}$, *with the first-order Taylor approximation,* $\max_{\mathbf{x}' \in B(\mathbf{x}, \delta_2)} ||\nabla_{\mathbf{x}} f(\mathbf{x}) - \nabla_{\mathbf{x}'} f(\mathbf{x}')||_2 \leq \delta_2 \max_i |\xi_i|$ *where* $\xi_i$ *is the* $i$-*th eigenvalue of the input Hessian* $\mathbf{H}_{\mathbf{x}}$.

**Smooth Surface Regularization**. Direct computation of the input Hessian can be expensive and in the case of ReLU networks, not possible to optimize as the second-order derivative is zero. Singla et al [40] introduce a closed-form solution of the input Hessian for ReLU networks and find its eigenvalues without doing exact engein-decomposition on the Hessian matrix. Motivated by Theorem 3 we propose Smooth Surface Regularization (SSR) to minimize the difference between Saliency Maps for nearby points.

**Proposition 3** (Singla et al' s Closed-form Formula for Input Hessian [40]). *Given a ReLU network* $f(\mathbf{x})$, *the input Hessian of the loss can be approximated by* $\tilde{\mathbf{H}}_{\mathbf{x}} = W(diag(\mathbf{p}) - \mathbf{p}^{\top}\mathbf{p})W^{\top}$, *where* $W$ *is the Jacobian matrix of the logits vector w.r.t to the input and* $\mathbf{p}$ *is the probits of the model.* $diag(\mathbf{p})$ *is an identity matrix with its diagonal replaced with* $\mathbf{p}$. $\tilde{\mathbf{H}}_{\mathbf{x}}$ *is positive semi-definite.*

**Definition 7** (Smooth Surface Regularization (SSR)). *Given data pairs* $(\mathbf{x}, y)$ *drawn from a distribution* $\mathcal{D}$, *the training objective of SSR is given by*

$$\min_{\boldsymbol{\theta}} \mathbb{E}_{(\mathbf{x},y) \sim \mathcal{D}} [\mathcal{L}((\mathbf{x}, y); \boldsymbol{\theta}) + \beta s \max_i \xi_i] \qquad (2)$$

*where* $\boldsymbol{\theta}$ *is the parameter vector of the the model* $f$, $\max_i \xi_i$ *is the largest eigenvalue of the Hessian matrix* $\tilde{\mathbf{H}}_{\mathbf{x}}$ *of the regular training loss* $\mathcal{L}$ *(e.g. Cross Entropy) w.r.t to the input.* $\beta$ *is a hyper-parameter for the penalty level and* $s$ *ensures the scale of the regularization term is comparable to regular loss.*

**An Alternative Stochastic Smoothing**. Users demanding robust interpretations are not always willing to afford extra budget for re-training. Except convolution with Guassian function, there is an alternative surface smoothing technique widely used in the mesh smoothing, *laplacian smoothing* [17, 42]. The key idea behind *laplacian smoothing* is that it replace the value of a point of interest on a surface using the aggregation of all neighbors within a specific distance. Adapting the motivation of *laplacian smoothing*, we examine the smoothing performance of UniGrad in this paper as well.

**Definition 8** (Uniform Gradient (UG) ). *Given an input point* $\mathbf{x}$, *the Uniform Gradient* $UG_c(\mathbf{x})$ *is defined as:* $g(\mathbf{x}) = \nabla_{\mathbf{x}} \mathbb{E}_p f(\mathbf{z}) = \mathbb{E}_p \nabla_{\mathbf{x}} f(\mathbf{z})$ *where* $p(\mathbf{z}) = U(\mathbf{x}, r)$ *is a uniform distribution centered at* $\mathbf{x}$ *with radius r.*

The second equality holds since the integral boundaries are not a function of the input due to the Leibniz integral rule. We include the visual comparisons of Uniform Gradient with other attribution methods in the Appendix D1. We also show the change of visualization against the noise radius $r$ in the Supplementary Material D2.

## 5   Experiments

In this section, we evaluate the performance of Attribution Attack on CIFAR-10 [21] and Flower [27] with ResNet-20 model and on ImageNet [11] with pre-trained ResNet-50 model. We apply the *top-k* [14] and *manipulate* attack [12] to each method. To evaluate the similarity between the original and perturbed attribution maps, except the metrics used by [14]: Top-k Intersection (k-in), Spearman's rank-order correlation (cor) and Mass Center Dislocation (cdl), we also include Cosine Distance (cosd) as higher cosine distance corresponds to higher $\ell_2$ distance between attribution maps and the value range is irrelevant with dimensions of input features, which provides comparable results across datasets compared to the $\ell_2$ distance. Further implementation details are included in the Supplementary Material C1.

### 5.1   Experiment I: Robustness via Stochastic Smoothing

In this experiment, we compare the robustness of different attribution methods on models with the natural training algorithm.

**Setup**. We optimize the Eq. (1) with maximum allowed perturbation $\epsilon_\infty = 2, 4, 8, 16$ for 500 images for CIFAR-10 and Flower and 1000 images for ImageNet. We first take the average scores over all

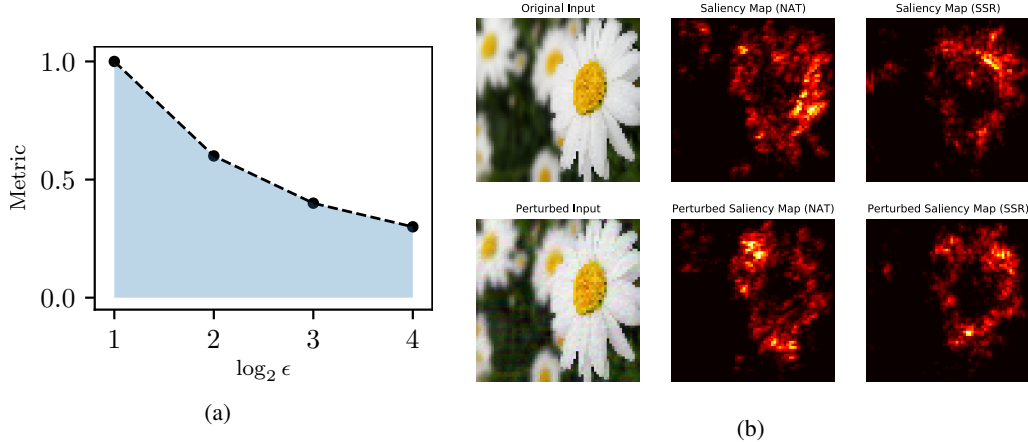

(a)

(b)

Figure 2: (a): Illustration of log-based AUC evaluation metric of Sec. 5. We evaluate each attack with $\epsilon_\infty = 2, 4, 8, 16$ and compute the area under the curve for each metric, e.g. top-k intersection. (b): An example of visual comparisons on the attribution attack on Saliency Map with nature training (NAT) and with Smooth Surface Regularization (SSR, $\beta = 0.3$) training. We then apply the *manipulate attack* with maximum allowed perturbation $\epsilon = 8$ in the $\ell_\infty$ space for 50 steps to the same image, respectively. The perturbed input for NAT model is omitted since they are visually similar.

evaluated images and then aggregate the results over different $\log_2 \epsilon_\infty$ using the area under the metric curve (AUC) (the $\log$ scale ensures each $\epsilon_\infty$ is equally treated and an illustration is shown in Fig. 2a). Higher AUC scores of k-in and cor or lower AUC scores of cdl and cosd indicate higher similarity between the original and the perturbed attribution maps. More information about hyper-parameters is included in the Supplementary Material B2.

**Numerical Analysis**. We show the reulst on cosd metric in Fig. 3 and the full results are included in Table 2 from Supplementary Material C shown. We conclude that attributions with stochastic smoothing, Smooth Gradient and Uniform Gradient, are showing better robustness than Saliency Map and Integrated Gradient on most metrics, espeically for ImageNet. What is more, Smooth Gradient does better on dataset with smaller sizes while Uniform does better on images with larger size. The role of dimensionality in stochastic smoothing and attribution robustness is an interesting future topic to research on.

## 5.2 Experiment II: Robustness via Regularization

Secondly, we evaluate the improvement of robustness via SSR. For the baseline methods, we include Madry's training [26] and IG-NORM [8], a recent proposed regularization to improve the robustness of Integrated Gradient.

**Setup**. SSR: we use the scaling coefficient $s = 1e6$ and the penalty $\beta = 0.3$. We discuss the choice of hyper-parameter for SSR in the the Supplementary Material B3. Madry's: we use cleverhans [31] implementation with PGD perturbation $\delta_2 = 0.25$ in the $\ell_2$ space and the number of PGD iterations equals to 30. IG-NORM: We use the author's release code with default penalty level $\gamma = 0.1$. We maintain the same training accuracies and record the per-epoch time with batch size of 32 on one NVIDIA Titan V. However, we discover that IG-NORM is sensitive to the weight initialization and the convergence rate is relatively slower than other in a significant way. We therefore present the best result among all attempts. For each attribution attack, we evaluate 500 images from CIFAR-10.

**Visualization**. We demonstrate a visual comparison between the perturbed Saliency maps of models with natural training and SSR training, respectively, in Fig 2b. After the same attack, regions with high density of attribution scores remain similar for the model with SSR training.

**Numerical Analysis**. We show the results on cosd in Fig. 4 and full experimental result is included in Table 3 of Supplementary Material C. We summarize the findings: 1) compared with results on the same model with natural training, SSR provides much better robustness nearly on all the metrics for all attribution methods; 2) SSR provides comparable and sometimes even better robustness compared

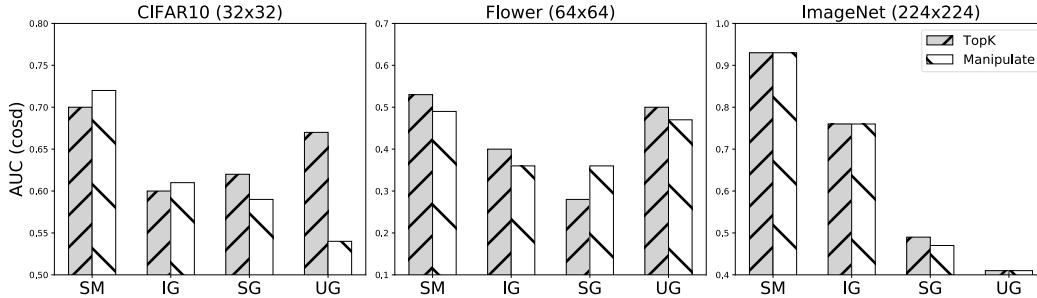

Figure 3: Evaluation with cosd of the top-k and manipulate attack on different dataset. We use $k = 20, 80$ and $1000$ pixels, respectively for CIFAR-10, Flower, and ImageNet to ensure the ratio of $k$ over the total number of pixels is approximately consistent across the dataset. Lower cosd indicates better attribution robustness. Each bar is computed by firstly taking the average scores over all evaluated images and then aggregating the results over different maximum allowed perturbation $\epsilon_\infty = 2, 4, 8, 16$ with the area under the metric curve (AUC). We include the full results on other metrics in the Table 2 from the Supplementary Material C.

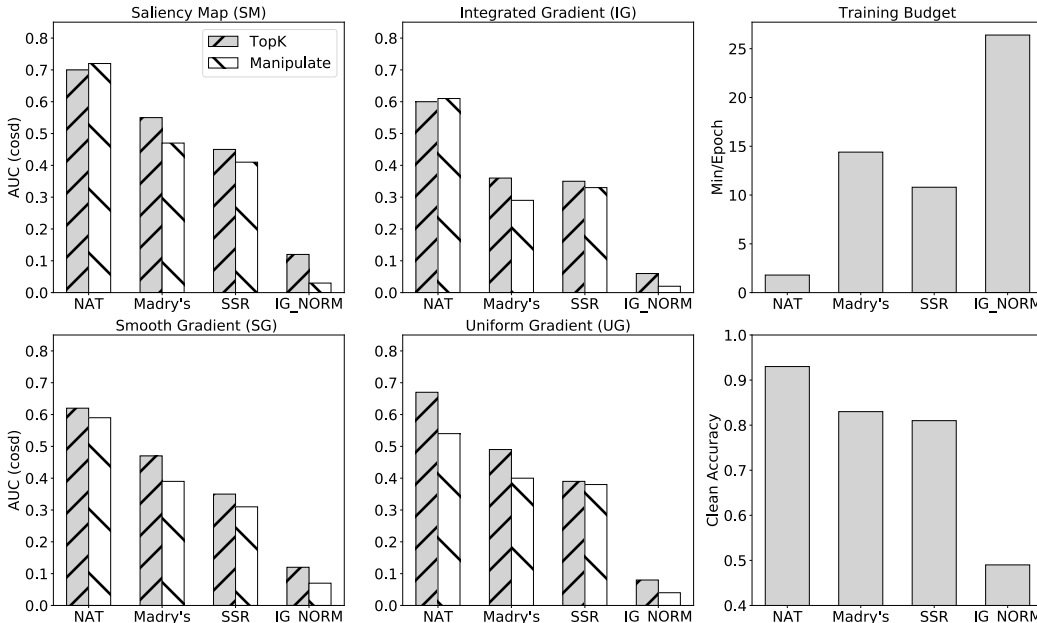

Figure 4: Evaluation with cosd of the top-k and manipulate attack on CIFAR-10 with different training algorithm. Lower cosd indicates better attribution robustness. The natural training is included in Table 2. We use $k = 20$. Each number in the table is computed by firstly taking the average scores over all evaluated images and then aggregating the results over different maximum allowed perturbation $\epsilon_\infty = 2, 4, 8, 16$ with the area under the metric curve shown in Fig. 2a. We include the full results on other metrics in the Table 3 in the Supplementary Material C.

to Madry's adversarial training. However, we also notice that playing with parameters in Madry's training influence robustness. We include the experiments of variability and parameter sweeping in Supplementary Material C. 3) Though IG-NORM provides the best performance, it has high costs of training time and accuracy, while SSR and Madry's training have lower costs.

### 5.3 Experiment III: Transferability

Given the nature of attribution attack is to find an adversarial example whose local geometry is significantly different from the input, there is a likelihood that an adversarial example of Saliency

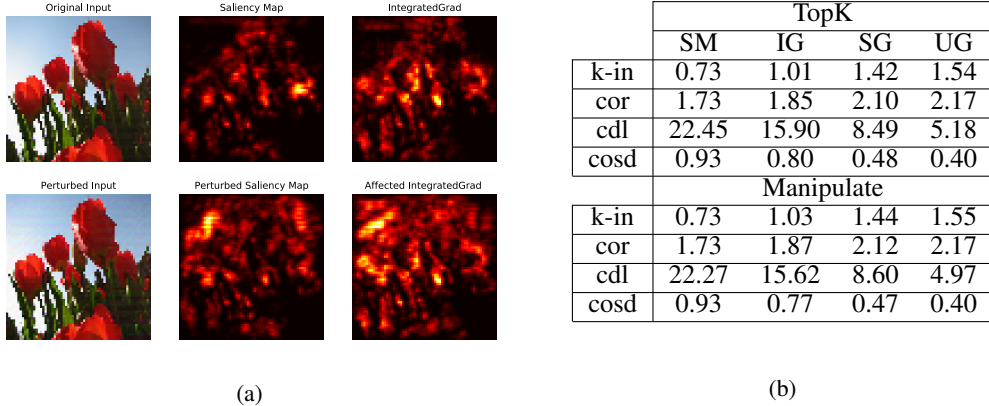

| | TopK | | | |
|---|---|---|---|---|
| | SM | IG | SG | UG |
| k-in | 0.73 | 1.01 | 1.42 | 1.54 |
| cor | 1.73 | 1.85 | 2.10 | 2.17 |
| cdl | 22.45 | 15.90 | 8.49 | 5.18 |
| cosd | 0.93 | 0.80 | 0.48 | 0.40 |
| | Manipulate | | | |
| k-in | 0.73 | 1.03 | 1.44 | 1.55 |
| cor | 1.73 | 1.87 | 2.12 | 2.17 |
| cdl | 22.27 | 15.62 | 8.60 | 4.97 |
| cosd | 0.93 | 0.77 | 0.47 | 0.40 |

(a)                                                    (b)

Figure 5: Transferability of attribution attacks. (a) Example manipulation attack on Saliency Map has similar effect on Integrated Gradient. (b) Effect on attribution maps under noise targeting Saliency Map *only*, average over 200 images from ImageNet on ResNet50 with standard training.

Map can be a valid attack to Integrated Gradient, which does not impose any surface smoothing technique to the local geometry. We verify this hypothesis by transferring the adversarial perturbation found on Saliency Map to other attribution maps. In Fig 5a, we compare the Integrated Gradient for the original and perturbed input, where the perturbation is targeted on Saliency Map with *manipulate attack* in a ball of $\delta_\infty = 4$. The result shows that the attack is successfully transferred from Saliency Map to Integrated Gradient. The experiment on 200 images shown in Fig 5b on ImageNet also shows that perturbation targeted on Saliency Map modifies Integrated Gradient in a more significant way than it does on Smooth Gradient or Uniform Gradient, which are motivated to smooth the local geometry. Therefore, empirical evaluations show that models with smoothed geometry have lower risk of being exposed to transfer attack.

## 6 Related Work and Discussion

**Characterization of Robustness**. In this paper, we characterize the local and global robustness of attributions with Lipchitz constants, where we find it similar to another metric SENS$_{\text{MAX}}$ [50]. The major difference between our work from Yeh et al. [50] is that we are motivated to measure the performance of attributions under the threat of an adversary where SENS$_{\text{MAX}}$ is motivated to evaluate the sensitivity of an attribution. As a result, Yeh et al. [50] employ Monte Carlo sampling to find SENS$_{\text{MAX}}$ of an attribution map; however, an adversarial example may not be easily sampled, especially in the high-dimensional input space, like ImageNet. Another line of work to characterize the robustness of attribution map is to measure the similarity between the attribution and the input by *alignment*, which is shown to be proportional to the robustness of model's prediction [13]. Towards robust attributions, Singh et al. [39] propose soft margin loss to improve the *alignment* for attributions.

**From Robust Attribution To Robust Prediction** Given a model with $(\lambda, \delta_2)$-local robust Saliency Map, $\lambda$ is proportional to $\max_{\mathbf{x}' \in B(\mathbf{x}, \delta_2)} ||\nabla_{\mathbf{x}} L - \nabla_{\mathbf{x}'} L||_2$ by Def. 6. Assume the maximization is achievable at $\mathbf{x}^*$, we have $\lambda \propto ||\nabla_{\mathbf{x}} L - \nabla_{\mathbf{x}^*} L||_2$. Triangle inequality offers $||\nabla_{\mathbf{x}} L - \nabla_{\mathbf{x}^*} L||_2 \leq ||\nabla_{\mathbf{x}} L||_2 + ||\nabla_{\mathbf{x}^*} L||_2 \leq 2||\nabla_{\mathbf{x}^\dagger} L||_2$ where $\mathbf{x}^\dagger = \arg\max_{\mathbf{x}' \in B(\mathbf{x}, \delta_2)} ||\nabla_{\mathbf{x}'} L||_2$. The above analysis shows that penalizing $\max_{\mathbf{x}' \in B(\mathbf{x}, \delta_2)} ||\nabla_{\mathbf{x}'} L||_2$ will lead to robust Saliency Map. We now show that the above penalty will also lead to robust prediction. Simon-Gabrie et al. [37] points out that training with $||\nabla_{\mathbf{x}} L||_p$ penalty is a first-order Taylor approximation to include the adversarial data (e.g. Madry's Training [26]) within the $\ell_q$ ball where $p, q$ are dual norms, which has been empirically discovered as well [29]. The conclusion above implies that penalizing the gradient norm at each training point leads to the increasing robustness of predictions. Obviously, the gradient norm $||\nabla_{\mathbf{x}} L||_p$ at each training point is upper-bounded by the local maximum norm $\max_{x' \in B(x, \delta_p)} ||\nabla_{\mathbf{x}} L||_p = ||\nabla_{\mathbf{x}^\dagger} L||_2$ (if choosing $p = 2$), the same target penalty term for training robust Saliency. Therefore, theoretically, improving the robustness of attribution can lead to increasing robustness of prediction. However, it is impossible for ReLU networks to run the inner-maximization due to the second-order derivatives being zeros, which is also the reason why we need Hessian approximation in this paper.

Alternatively, one can replace ReLU with other second-order differentiable modules, e.g. Softplus, to run the min-max optimization, the intensive computation is probably still not generally affordable across the entire ML community. We include full discussion in the Supplementary Material B.

**From Robust Prediction To Robust Attribution.** In ReLU networks, provable robust predictions may lead to robust attributions, e.g. MMR [10]. One benefit of ReLU networks is that the model behaves linearly within each activation polytope [18]; therefore, Saliency Map within each polytobe becomes consistent. MMR increases the size of local linear regions by pushing the boundaries of activate polytobes away from each training point such that the model is locally linear in a bigger epsilon ball compared to the natural training. However, provable defenses against prediction attacks currently suffer from the lack of scalability to large models.

**Surface Smoothing**. Smoothing the geometry with Hessians has been widely adopted for different purpose. We use Singla et al' s Closed-form Formula given that it is the first approach that returns Hessian approximation for ReLU networks [40]. There are other approaches that approximate the Hessian's spectrum norm, e.g. by using Frobenius norm and the finite difference of Hessian-vector product [28] and by regularizing the spectrum norms of weights [51].

**Other Post-hoc Approach.** An ad-hoc regularization can be an extra budget for people with pre-trained models. Except Smooth Gradient and Uniform Gradient discussed in the paper, Levine et al. [24] propose Sparsified SmoothGrad as a certified robust version of Smooth Gradient.

# 7    Conclusion

We demonstrated that lack of robustness for gradient-based attribution methods can be characterized by Lipschitz continuity or smooth geometry, e.g. Smooth Gradient is more robust than Saliency Map and Integrated Gradient, both theoretically and empirically. We proposed *Smooth Surface Regularization* to improve the robustness of all gradient-based attribution methods. The method is more efficient than existing output (Mądry's training) and attribution robustness (IG-Norm) approaches, and applies to networks with ReLU. We exemplified smoothing with *Uniform Gradient*, a variant of Smooth Gradient with better robustness for some similarity metrics than Smooth Gradient while neither form of smoothing achieves best performance overall. This indicates future directions to investigate ideal smoothing parameters. Our methods can be used both for training models robust in attribution (SSR) and for robustly explaining existing pre-trained models (UG). These tools extend a practitioner's transparency and explainability toolkit invaluable in especially high-stakes applications.

## Acknowledgements

This work was developed with the support of NSF grant CNS-1704845 as well as by DARPA and the Air Force Research Laboratory under agreement number FA8750-15-2-0277. The U.S. Government is authorized to reproduce and distribute reprints for Governmental purposes not withstanding any copyright notation thereon. The views, opinions, and/or findings expressed are those of the author(s) and should not be interpreted as representing the official views or policies of DARPA, the Air Force Research Laboratory, the National Science Foundation, or the U.S. Government.

## Broader Impact

Our work is expected to have general positive broader impacts on the uses of machine learning in the broader society. Specifically, we are addressing the continual lack of transparency in deep learning and the potential of intentional abuse of systems employing deep learning. We hope that work such as ours will be used to build more trustworthy systems and make them more resilient to adversarial influence. As the impact of deep learning in general grows, so will the impact of transparency research such as ours. Depending on the use cases, such as work on algorithmic fairness, transparency tools such as hours can have positive impact on disadvantaged groups who either enjoy reduced benefits of machine learning or are susceptible to unfair decisioning from them. While any work in an adversarial setting can be misused as an instruction manual for defeating or subverting the proposed or similar methods, we believe the publication of the work is more directly useful in ways positive to the broader society.

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
