[Supplementary Material]

# Supplementary Material

## Supplymentary Material A

### A1. Proof of Theorm 1

**Theorm 1** *Given a model $f(\mathbf{x})$ is $(L, \delta_2)$-locally lipchitz continious in a ball $B(\mathbf{x}, \delta_2)$, then the Saliency Map is $(\lambda, \delta_2)$-local robustenss where $\lambda$ is $\mathcal{O}(L)$.*

*Proof.* We first introduce the following lemma.

**Lemma 1** (Lipschitz continuity and gradient norm [32]). *If a general function $h : \mathbb{R}^d \to \mathbb{R}$ is L-locally lipchitz continuous and continuously first-order differentiable in $B(\mathbf{x}, \delta_p)$, then*

$$L = \max_{\mathbf{x}' \in B(\mathbf{x}, \delta_p)} ||\nabla_{\mathbf{x}'} h(\mathbf{x}')||_q \tag{3}$$

*where $\frac{1}{p} + \frac{1}{q} = 1, 1 \leq p, q, \leq \infty$.*

We start to prove Theorem 1. By Def. 6, we write the robustness of Saliency Map as

$$\lambda = \max_{\mathbf{x}' \in B} \frac{||\nabla_{\mathbf{x}} f(\mathbf{x}) - \nabla_{\mathbf{x}'} f(\mathbf{x}')||_2}{||\mathbf{x} - \mathbf{x}'||_2} \tag{4}$$

Assume $x^* = \arg\max_{\mathbf{x}' \in B} \frac{||\nabla_{\mathbf{x}} f(\mathbf{x}) - \nabla_{\mathbf{x}'} f(\mathbf{x}')||_2}{||\mathbf{x} - \mathbf{x}'||_2}$ therefore

$$\lambda = \frac{||\nabla_{\mathbf{x}} f(\mathbf{x}) - \nabla_{\mathbf{x}^*} f(\mathbf{x}^*))||_2}{||\mathbf{x} - \mathbf{x}^*||_2} \leq \frac{||\nabla_{\mathbf{x}} f(\mathbf{x})||_2 + ||\nabla_{\mathbf{x}^*} f(\mathbf{x}^*))||_2}{||\mathbf{x} - \mathbf{x}^*||_2} \leq \frac{2||\nabla_{\mathbf{x}^\dagger} f(\mathbf{x}^\dagger)||_2}{||\mathbf{x} - \mathbf{x}^*||_2} \tag{5}$$

where $\mathbf{x}^\dagger = \arg\max_{\mathbf{x}' \in B(\mathbf{x}, \delta_2)} ||\nabla_{\mathbf{x}'} f(\mathbf{x}')||_2$. Since $f(\mathbf{x})$ is $(L, \delta_2)$-locally lipchitz continious, with Lemma 1 and by choosing $p = 2$, we have $L = ||\nabla_{\mathbf{x}^\dagger} f(\mathbf{x}^\dagger)||_2$.

Therefore, we end the proof with

$$\lambda \leq \frac{2L}{||\mathbf{x} - \mathbf{x}^*||_2} \tag{6}$$

which indicates $\lambda$ is proportional to $L$. $\square$

### A2. Proof of Proposition 1

**Proposition 1** *Given a model $f(\mathbf{x})$ and we assume $F = \max_{\mathbf{x} \in B} |f(\mathbf{x})| < \infty$, and a user-defined noise level $\sigma$, the following equation holds for SmoothGrad: $g(\mathbf{x}) = \mathbb{E}_{z \sim \mathcal{N}(\mathbf{x}, \sigma^2 \mathbf{I})} \nabla_{\mathbf{z}} f(\mathbf{z}) = \nabla_{\mathbf{x}}[(f * q)(\mathbf{x})]$ where $q(\mathbf{x}) \sim \mathcal{N}(\mathbf{0}, \sigma^2 \mathbf{I})$ and $*$ denotes the convolution.*

*Proof.* The proof of Proposition 1 follows Bonnets' Theorem and Stein's Theorem which have been discussed by Lin et al. [25]. We first show that

**Lemma 2.** *l Given a locally Lipschitz continuous function $f : \mathbb{R}^d \to \mathbb{R}$, and a Gaussian distribution $q(\mathbf{z}) \sim \mathcal{N}(\mathbf{x}, \sigma^2 \mathbf{I})$, we have*

$$\mathbb{E}_q[\nabla_{\mathbf{z}} h(\mathbf{z})] = \mathbb{E}_q[(\sigma^2 \mathbf{I})^{-1}(\mathbf{z} - \mathbf{x}) h(\mathbf{z})] \tag{7}$$

The proof of Lemma 2 is described by the Lemma 5 of Lin et al. [25] and follow the proof of Theorem 3 of Lin et al. [25] we show that

$$\nabla_{\mathbf{x}} \mathbb{E}_q[h(\mathbf{z})] = \int h(\mathbf{z}) \nabla_{\mathbf{x}} \mathcal{N}(\mathbf{z}|\mathbf{x}, \sigma^2 \mathbf{I}) d\mathbf{z} \tag{8}$$

$$= \int h(\mathbf{z})(\sigma^2 \mathbf{I})^{-1}(\mathbf{z} - \mathbf{x}) \mathcal{N}(\mathbf{z}|\mathbf{x}, \sigma^2 \mathbf{I}) d\mathbf{z} \tag{9}$$

$$= \mathbb{E}_q[(\sigma^2 \mathbf{I})^{-1}(\mathbf{z} - \mathbf{x}) h(\mathbf{z})] \tag{10}$$

Therefore, we have $\mathbb{E}_q[\nabla_{\mathbf{z}} h(\mathbf{z})] = \nabla_{\mathbf{x}} \mathbb{E}_q[h(\mathbf{z})]$. Since

$$\mathbb{E}_q[h(\mathbf{z})] = \int h(\mathbf{z}) \mathcal{N}(\mathbf{z}|\mathbf{x}, \sigma^2 \mathbf{I}) d\mathbf{z} \tag{11}$$

$$= \int h(\mathbf{z}) q(\mathbf{z} - \mathbf{x}) d\mathbf{z} \tag{12}$$

$$= \int h(\mathbf{z}) q(\mathbf{x} - \mathbf{z}) d\mathbf{z} \tag{13}$$

By the definition of convolution, $\mathbb{E}_q[h(\mathbf{z})] = f * q$. Hence, we prove the proposition 1.

$\square$

### A3. Proof of Theorem 2

**Theorem 2** *Given a model $f(\mathbf{x})$ and we assume $|f(\mathbf{x})| < F < \infty$ in the input space, Smooth Gradient with a user-defined noise level $\sigma$ is $\lambda$-robustness globally where $\lambda \leq 2F/\sigma^2$*

To prove Theorem 2, we first introduce the following lemmas.

**Lemma 3.** *The input Hessian $\tilde{\mathbf{H}}_{\mathbf{x}}$ of $\tilde{f}_\sigma(\mathbf{x})$ is given by $\tilde{\mathbf{H}}_{\mathbf{x}} = \frac{1}{\sigma^4} \mathbb{E}_p[(\mathbf{z} - \mathbf{x})(\mathbf{z} - \mathbf{x})^\top - \sigma^2 \mathbf{I}) f(\mathbf{z})]$*

*Proof.* The first-order derivative of $\tilde{f}_\sigma(\mathbf{x})$ has been offered by Lemma 2, with the help of which, we find the second-order derivatives following the proof in [53].

$\square$

We then generalize Lemma 1 to a multivariate output function.

**Lemma 4.** *If a general function $h : \mathbb{R}^d \to \mathbb{R}^m$ is L-locally lipchitz continuous measured in $\ell_2$ space and continuously first-order differentiable in $B(\mathbf{x}, \delta_p)$, then*

$$L = \max_{\mathbf{x}' \in B(\mathbf{x}, \delta_p)} ||\nabla_{\mathbf{x}'} h(\mathbf{x}')||_* \tag{14}$$

*where $||\mathbf{M}||_* = \sup_{\{||\mathbf{x}||_2 = 1, \mathbf{x} \in \mathbb{R}^d\}} ||\mathbf{M}\mathbf{x}||_2$ is the spectral norm for arbitrary matrix $\mathbf{M} \in \mathbb{R}^{m \times d}$.*

The proof of Lemma 4 is omitted since it is the repeat of Virmaux et al's [48] Theorem 1.

We now prove Theorem 2. The robustness coefficient $\tilde{\lambda}$ of SmoothGrad is equivalent to the robustness of Saliency Map on $\tilde{f}_\sigma(\mathbf{x})$ based on Lemma 1. By Def. 6, the robustness of Saliency Map is just the local lipschitz smoothness constant, in the other word, the local lipschitz continuity constant of $\nabla_{\mathbf{x}} \tilde{f}_\sigma(\mathbf{x})$. Subsitute the $h(\mathbf{x})$ in Lemma 4 with $\nabla_{\mathbf{x}} \tilde{f}_\sigma(\mathbf{x})$, we have

$$\tilde{\lambda} = \max_{\mathbf{x}' \in B(\mathbf{x}, \delta_p)} ||\tilde{\mathbf{H}}_{\mathbf{x}'}||_* \tag{15}$$

Let $\mathbf{x}^\dagger = \arg\max_{\mathbf{x}' \in B(\mathbf{x}, \delta_p)} ||\tilde{\mathbf{H}}_{\mathbf{x}'}||_*$,

$$\tilde{\lambda} = ||\tilde{\mathbf{H}}_{\mathbf{x}^\dagger}||_* = ||\frac{1}{\sigma^4} \mathbb{E}_p[(\mathbf{z} - \mathbf{x}^\dagger)(\mathbf{z} - \mathbf{x}^\dagger)^\top - \sigma^2 \mathbf{I}) f(\mathbf{z})]||_* \quad \text{(Lemma 3)} \tag{16}$$

$$\leq \frac{1}{\sigma^4} \{||\mathbb{E}_p[(\mathbf{z} - \mathbf{x}^\dagger)(\mathbf{z} - \mathbf{x}^\dagger)^\top f(\mathbf{z})]||_* + \sigma^2 ||\mathbb{E}_p[f(\mathbf{z})\mathbf{I}]||_*\} \tag{17}$$

$$\leq \frac{1}{\sigma^4} \{||\mathbb{E}_p[(\mathbf{z} - \mathbf{x}^\dagger)(\mathbf{z} - \mathbf{x}^\dagger)^\top |f(\mathbf{z})|]||_* + \sigma^2 ||\mathbb{E}_p[|f(\mathbf{z})|\mathbf{I}]||_*\} \tag{18}$$

$$\leq \frac{1}{\sigma^4} (F\sigma^2 + \sigma^2 F) \tag{19}$$

$$\tilde{\lambda} \leq \frac{2F}{\sigma^2} \tag{20}$$

## A4: Proof of Proposition 2

**Proposition** 2. *Given a model $f(\mathbf{x})$ is $(L, \delta_2)$-locally lipchitz continuous in the ball $B(\mathbf{x}, \delta_p)$ and assuming $\sup_{\mathbf{x} \in \mathbb{R}^d} |f(\mathbf{x})| = F < \infty$. With a proper chosen noise level $\sigma > \sqrt{\delta_2 F/L}$, the upper-bound of the local robustness of Smooth Gradient is always smaller than the upper-bound of the local robustness of Saliency Map.*

*Proof.* Assume Smooth Gradient is $(\lambda', \delta_2)$-local robust and $\tilde{\lambda}$-global robust. Denote $B$. Since the global lipchitz constant is greater or equal to the local lipschitz constant [4] and by definition local robustness is the local lipchitz constant for the attribution map, we have

$$\lambda' \leq \tilde{\lambda} \leq \frac{2F}{\sigma^2} \tag{21}$$

The second inequality is based on Theorem 2. We now start to prove the proposition. Given $\sigma > 0, \sqrt{\delta_2 F/L} > 0$, we have

$$\sigma^2 > \frac{F}{L}\delta_2 \tag{22}$$

We consider a maximizer $\mathbf{x}^* \in B(\mathbf{x}, \delta_p)$ such that

$$\mathbf{x}^* = \arg \max_{\mathbf{x}' \in B(\mathbf{x}, \delta_2)} \frac{|f(\mathbf{x}) - f(\mathbf{x}')|}{||\mathbf{x} - \mathbf{x}'||_2} \tag{23}$$

Since $||\mathbf{x} - \mathbf{x}'|| \leq \delta_2$ for any $\mathbf{x}' \in B(\mathbf{x}, \delta_2)$, we have $||\mathbf{x} - \mathbf{x}^*||_2 \leq \delta_2$ We plug in this into Eq. (22)

$$\sigma^2 > \frac{||\mathbf{x} - \mathbf{x}^*||_2}{L}F \tag{24}$$

$$\frac{F}{\sigma^2} \leq \frac{L}{||\mathbf{x} - \mathbf{x}^*||_2} \tag{25}$$

$$\frac{2F}{\sigma^2} \leq \frac{2L}{||\mathbf{x} - \mathbf{x}^*||_2} \tag{26}$$

Based on Equation (6) in the proof of Theorem 1, given a model $f(\mathbf{x})$ is $(L, \delta_2)$-locally lipchitz continuous in the ball $B(\mathbf{x}, \delta_p)$, Saliency Map is $(\lambda, \delta_2)$ robustness and $\lambda \leq \frac{2L}{||\mathbf{x} - \mathbf{x}^*||_2}$ and Thereom 2, we prove the proporstion. $\qquad \square$

## A5. Proof of Theorem 3

**Theorem 3** *Given a twice-differentiable function $f : \mathbb{R}^{d_1} \to \mathbb{R}^{d_2}$, with the first-order Taylor approximation, $\max_{\mathbf{x}' \in B(\mathbf{x}, \delta_2)} ||\nabla_{\mathbf{x}} f(\mathbf{x}) - \nabla_{\mathbf{x}'} f(\mathbf{x}')||_2 \leq \delta_2 \max_i |\xi_i|$ where $\xi_i$ is the $i$-th eigenvalue of the input Hessian $\mathbf{H}_{\mathbf{x}}$.*

*Proof.*

$$\max_{\mathbf{x}' \in B(\mathbf{x}, \delta_2)} ||\nabla_{\mathbf{x}} f(\mathbf{x}) - \nabla_{\mathbf{x}'} f(\mathbf{x}')||_2 \approx \max_{\mathbf{x}' \in B(\mathbf{x}, \delta_2)} ||\mathbf{H}_{\mathbf{x}}(\mathbf{x}' - \mathbf{x})||_2 \quad \text{(Taylor Expansion)} \tag{27}$$

$$= \max_{\mathbf{x}' \in B(\mathbf{x}, \delta_2)} ||\mathbf{H}_{\mathbf{x}}||\mathbf{x}' - \mathbf{x}||_2 \frac{\mathbf{x}' - \mathbf{x}}{||\mathbf{x}' - \mathbf{x}||_2}||_2 \tag{28}$$

$$\leq \max_{\mathbf{x}' \in B(\mathbf{x}, \delta_2)} ||\mathbf{H}_{\mathbf{x}}\delta_2 \frac{\mathbf{x}' - \mathbf{x}}{||\mathbf{x}' - \mathbf{x}||_2}||_2 \tag{29}$$

$$= \max_{||\boldsymbol{\epsilon}||_2 = 1} \delta_2 ||\mathbf{H}_{\mathbf{x}}\boldsymbol{\epsilon}||_2 \quad \text{(Definition of Spectral Norm)} \tag{30}$$

$$= \delta_2 \max_i |\xi_i| \tag{31}$$

$$\tag{32}$$

$$\square$$

## A6. Proof of Proposition 3

**Proposition 3**(Singla et al' s Closed-form Formula for Input Hessian) *Given a ReLU network $f(\mathbf{x})$, the input Hessian of the loss can be approximated by $\tilde{\mathbf{H}}_\mathbf{x} = W(diag(\mathbf{p}) - \mathbf{p}^\top \mathbf{p})W^\top$, where $W$ is the Jacobian matrix of the logits vector w.r.t to the input and $\mathbf{p}$ is the probits of the model. $diag(\mathbf{p})$ is an identity matrix with its diagonal replaced with $\mathbf{p}$. $\tilde{\mathbf{H}}_\mathbf{x}$ is positive semi-definite.*

*Proof.* The proof of Proposition 3 follows the proof of proposition 1 and Theorem 2 of Singla et al [40].

**Extra Notation** Let $\mathbf{W}_l$ be the weight matrix of $l$-th layer to the $l + 1$-th layer and $\mathbf{b}_l$ be the bias of the $l$-th layer. Denote the ReLU activation as $\sigma(\cdot)$. We use $\hat{\mathbf{y}}$, $\mathbf{p}$ and $\mathbf{y}$ to represent the pre-softmax output, the pose-softmax probability distribution and the one-hot ground-truth label.

Firstly, we describe the local-linearity of a ReLU network. The pre-softmax output $\hat{\mathbf{y}}$ of a ReLU network can be written as

$$\hat{\mathbf{y}} = \sigma(\cdots\sigma(\mathbf{W}_2^\top \sigma(\mathbf{W}_1^\top \mathbf{x} + \mathbf{b}_1) + \mathbf{b}_2)\cdots) \tag{33}$$

$$= \mathbf{W}^\top \mathbf{x} + \mathbf{b} \tag{34}$$

where each column $\mathbf{W}_i = \frac{\partial \hat{\mathbf{y}}_i}{\partial \mathbf{x}}$. Therefore, assume we use Cross-Entropy loss as the training loss, we can write the first-order gradient of the loss w.r.t input as

$$\nabla_\mathbf{x}\mathcal{L} = \frac{\partial \mathcal{L}}{\partial \hat{\mathbf{y}}}\frac{\partial \hat{\mathbf{y}}}{\mathbf{x}} = \mathbf{W}(\mathbf{p} - \mathbf{y}) \tag{35}$$

Thirdly, we compute the input Hessian

$$\mathbf{H}_\mathbf{x} = \nabla_\mathbf{x}^2\mathcal{L} = \nabla_\mathbf{x}(\mathbf{W}(\mathbf{p} - \mathbf{y})) = \nabla_\mathbf{x}(\sum_i \mathbf{W}_i(\mathbf{p}_i - \mathbf{y}_i)) \tag{36}$$

$$= \sum_i \mathbf{W}_i \nabla_\mathbf{x}(\mathbf{p}_i - \mathbf{y}_i) \tag{37}$$

$$= \sum_i \mathbf{W}_i (\nabla_\mathbf{x}\mathbf{p}_i)^\top \tag{38}$$

$$= \sum_i \mathbf{W}_i (\sum_j \frac{\partial \mathbf{p}_j}{\partial \hat{\mathbf{y}}_j}\frac{\partial \hat{\mathbf{y}}_j}{\partial \mathbf{x}_j})^\top \tag{39}$$

$$= \sum_i \mathbf{W}_i (\sum_j \frac{\partial \mathbf{p}_j}{\partial \hat{\mathbf{y}}_j}\mathbf{W}_j)^\top \tag{40}$$

$$= \sum_i \sum_j \mathbf{W}_i \frac{\partial \mathbf{p}_j}{\partial \hat{\mathbf{y}}_j}^\top \mathbf{W}_j^\top \tag{41}$$

$$= \mathbf{WAW}^\top \tag{42}$$

where $\mathbf{A} = \frac{\partial \mathbf{p}_j}{\partial \hat{\mathbf{y}}_j}^\top = \text{diag}(\mathbf{p}) - \mathbf{pp}^\top$ and $\text{diag}(\mathbf{p})$ denotes a matrix with $\mathbf{p}$ as the diagonal entries and 0 otherwise. We use $\tilde{\mathbf{H}}_\mathbf{x}$ to denote $\mathbf{WAW}^\top$, the input Hessian of the Cross-Entropy loss. Finally, we show $\tilde{\mathbf{H}}_\mathbf{x}$ is semi-positive definite (PSD). The basic idea is to show $\mathbf{A}$ is PSD and using Cholesky decompostion we show $\tilde{\mathbf{H}}_\mathbf{x}$ is PSD as well. To show $\mathbf{A}$ is PSD, we consider

$$\sum_{j\neq i}|\mathbf{A}_{ij}| = \sum_{j\neq i}|-\mathbf{p}_i\mathbf{p}_j| = \mathbf{p}_i\sum_{j\neq i}\mathbf{p}_j = \mathbf{p}_i(1 - \mathbf{p}_i) > 0 \tag{43}$$

The diagonal entries $|\mathbf{A}_{ij}| = \mathbf{p}_i(1 - \mathbf{p}_i)$. With Gershgorim Circle theorem, all eigenvalues of $\mathbf{A}$ is positive, so $\mathbf{A}$ is PSD. Therefore, we can find Cholesky decomposition of $\mathbf{A}$ such that $\mathbf{A} = \mathbf{MM}^\top$. Then $\tilde{\mathbf{H}}_\mathbf{x} = \mathbf{WMM}^\top\mathbf{W}^\top = \mathbf{WM}(\mathbf{WM})^\top$, which means the Cholesky decomposition of $\tilde{\mathbf{H}}_\mathbf{x}$ exists, so $\tilde{\mathbf{H}}_\mathbf{x}$ is PSD as well.

$\square$

## Supplementary Material B

### B1. Eigenvalue Computation of $\tilde{\mathbf{H}}_{\mathbf{x}}$

In this section, we discuss the computation of eigenvalues of the input Hessian in SSR defined in Def. 7. As pointed out by Supplymentary Material A6, we can write

$$\tilde{\mathbf{H}}_{\mathbf{x}} = \mathbf{WM}(\mathbf{WM})^{\top} = \mathbf{BB}^{\top} \qquad (44)$$

where $\mathbf{B} = \mathbf{WM}$. Let the SVD of $\mathbf{B}$ be $\mathbf{U\Sigma V}$, so that

$$\mathbf{BB}^{\top} = \mathbf{U\Sigma}^2\mathbf{U}^{\top} \qquad (45)$$

Note that $\mathbf{B}^{\top}\mathbf{B} = \mathbf{V\Sigma}^2\mathbf{V}^{\top}$ whose singular values are identical to $\tilde{\mathbf{H}}_{\mathbf{x}}$ and the dimension of $\mathbf{B}^{\top}\mathbf{B}$ is $c \times c$ where $c$ is the number of classes. Given $\tilde{\mathbf{H}}_{\mathbf{x}}$ is PSD, so the singular values and eigenvalues coincide. Therefore, we can compute the eigenvalues of $\mathbf{B}^{\top}\mathbf{B}$ instead of running eigen-decomposition on $\tilde{\mathbf{H}}_{\mathbf{x}}$ directly.

### B2. Adversarial training and robustness of attribution

In the Sec. 4 we discuss the connection between the robustness of prediction and the robustness of attributions with $\max_{\mathbf{x}' \in B(\mathbf{x},\delta_2)} ||\nabla_{\mathbf{x}}\mathcal{L}||_2$ with the first-order Talyer Expansion. In this section, we also look into the second-order term. Firstly, we denote $\Delta\mathcal{L} = \max_{||\boldsymbol{\epsilon}|| \leq \delta_2} |\mathcal{L}((\mathbf{x}+\boldsymbol{\epsilon}, y) - \mathcal{L}((\mathbf{x}, y)|$. With the second-order Taylor expansion, we can write

$$\mathcal{L}(\mathbf{x}+\boldsymbol{\epsilon}, y) \approx \mathcal{L} + \nabla_{\mathbf{x}}\mathcal{L}^{\top}\boldsymbol{\epsilon} + \frac{1}{2}\boldsymbol{\epsilon}^{\top}\mathbf{H}_{\mathbf{x}}\boldsymbol{\epsilon} \qquad (46)$$

Therefore, we have

$$\Delta\mathcal{L} \approx \max_{||\boldsymbol{\epsilon}|| \leq \delta_2} |\nabla_{\mathbf{x}}\mathcal{L}^{\top}\boldsymbol{\epsilon} + \frac{1}{2}\boldsymbol{\epsilon}^{\top}\mathbf{H}_{\mathbf{x}}\boldsymbol{\epsilon}| \qquad (47)$$

$$\leq \max_{||\boldsymbol{\epsilon}|| \leq \delta_2} |\nabla_{\mathbf{x}}\mathcal{L}^{\top}\boldsymbol{\epsilon}| + \max_{||\boldsymbol{\epsilon}|| \leq \delta_2} |\frac{1}{2}\boldsymbol{\epsilon}^{\top}\mathbf{H}_{\mathbf{x}}\boldsymbol{\epsilon}| \qquad (48)$$

Simon-Gabriel et al. [37] demonstrates that the first term $\max_{||\boldsymbol{\epsilon}|| \leq \delta_2} |\nabla_{\mathbf{x}}\mathcal{L}^{\top}\boldsymbol{\epsilon}| = \delta_2||\nabla_{\mathbf{x}}\mathcal{L}||_2$, we now focus on the second term

$$\max_{||\boldsymbol{\epsilon}|| \leq \delta_2} |\frac{1}{2}\boldsymbol{\epsilon}^{\top}\mathbf{H}_{\mathbf{x}}\boldsymbol{\epsilon}| = \max_{||\boldsymbol{\epsilon}|| \leq \delta_2} \frac{||\boldsymbol{\epsilon}||_2^2}{2} |\frac{\boldsymbol{\epsilon}^{\top}}{||\boldsymbol{\epsilon}||_2}\mathbf{H}_{\mathbf{x}}\frac{\boldsymbol{\epsilon}}{||\boldsymbol{\epsilon}||_2}| \qquad (49)$$

$$\leq \frac{\delta_2^2}{2} \max_{||\boldsymbol{\epsilon}||=1} |\boldsymbol{\epsilon}^{\top}\mathbf{H}_{\mathbf{x}}\boldsymbol{\epsilon}| \qquad (50)$$

With Rayleigh quotient, we have $\max_{||\boldsymbol{\epsilon}||=1} |\boldsymbol{\epsilon}^{\top}\mathbf{H}_{\mathbf{x}}\boldsymbol{\epsilon}| = \max_i |\xi_i|$ where $\xi_i$ is the $i$-th eigenvalue of the Hessian. However, as mentioned by Simon-Gabriel et al. [37], the higher-order term has very limited contribution to the upper-bound of $\Delta\mathcal{L}$ compared with the first-order term. Therefore, with second-order Taylor's approximation, adversarial training optimizes a lower-bound of the sum of the gradient norm and the largest eigenvalue of input hessian.

## Supplementary Material C

### C1. Implementation Details for Experiments

**Attribution Methods.** We use the predicted class as the quantity of interest for all attribution methods. For IG, SG, and UG, we use 50 samples to approximate the expectation. For IG, we use the zero baseline input. For SG, we use the noise standard deviation $\sigma = \texttt{ratio} \times (u_x - l_x)$ where $u_x$ is the maximum pixel value of the input and $l_x$ is the minimum pixel value of the input and the noise ratio $\texttt{ratio} = 0.1$ for CIFAR and Flower, $\texttt{ratio} = 0.2$ for ImageNet. For UG, we use $r = 4$ as the noise radius for data range of [0, 255] in CIFAR and Flower and $r = 0.2 \times (u_x - l_x)$ for ImageNet. We

employ bigger noise levels in ImageNet for both SG and UG because empirically we find higher noise levels produce better visualizations (see Fig. 10) in dataset with such high dimensions.

**Evaluation Metrics.** Let $\mathbf{z}$ and $\mathbf{z}'$ be the original and perturbed attribution maps, respectively, attribution attacks are evaluated with following metrics:

- **Top-k Intersection (k-in)** measures the intersection between features with top-k attribution scores in the original and perturbed attribution map: $\sum_{i \in K} \mathbf{n}(\mathbf{z}')_i$ where $\mathbf{n}(\mathbf{x}) = |\mathbf{x}| / \sum_j^d |\mathbf{x}_j|$ and $K$ is the set of k-largest dimensions of $\mathbf{n}(\mathbf{z})$ [14].

- **Spearman's rank-order correlation (cor)** [43] compares the rank orders of $\mathbf{z}$ and $\mathbf{z}'$ as features with higher rank in the attribution map are often interpreted as more important.[1]

- **Mass Center Dislocation (cdl)** measures the spatial displacement of the "center" of attribution scores by $\sum_i^d [\mathbf{z}_i - \mathbf{z}'_i] i$ [14].

- **Cosine Distance (cosd)** measures the change of directions bewteen attribution maps by $1 - \langle \mathbf{z}, \mathbf{z}' \rangle / ||\mathbf{z}||_2 ||\mathbf{z}'||_2$.

**Attribution Attacks.** To implement the attribution attack, we adapt the release code[2] by [14] and we make the following changes:

1. We change the clipping function to projection to bound the norm of the total perturbation.
2. We use `grad × input` for Saliency Map, Smooth Gradient, and Uniform Gradient.
3. For the manipulate attack, we use the default parameters $\beta_0 = 1e11$ and $\beta_1 = 1e6$ in the original paper.

For all experiments and all attribution attacks, we run the attack for 50 iterations.

**IG-NORM Regularization.** We use the following parameters to run IG-NORM regularization which are default parameters in the release code[3]. We use Adam in the training.

| epochs | batch_size | $\epsilon_\infty$ | $\gamma$ | nbiter | m | approx_factor | step_size |
|--------|-----------|-------------------|----------|--------|-----|---------------|-----------|
| 50 | 16 | 8/255 | 0.1 | 7 | 50 | 10 | 2/255 |

Table 1: Hyper-parameters used in IG-NORM training

where

- `epochs`: the number of epochs in the training.

- `batch_size`: size of each mini-batch in the gradient descent.

- $\epsilon_\infty$: maximum allowed perturbation of the the input to run the inner-maximization

- $\gamma$: penalty level of the IG loss.

- `nbiter`: the number of iterations used to approximate the inner-maximization of the IG-loss

- `m`: the number of samples used to approximate the path integral of IG.

- `approx_factor`: the actual samples used to approximate IG is `m/approx_factor`

- `step_size`: the size of each PGD iteration.

| | | TopK Attack | | | | Manipulate Attack | | | |
|---|---|---|---|---|---|---|---|---|---|
| | | SM | IG | SG | UG | SM | IG | SG | UG |
| CIFAR-10 32 × 32 | k-in | 0.51 | 0.84 | 0.75 | **2.94** | 0.49 | 0.81 | 0.85 | **2.95** |
| | cor | 1.85 | **1.96** | 1.67 | 1.85 | 1.82 | **1.95** | 1.70 | 1.89 |
| | cdl | 2.93 | 3.05 | **1.97** | 3.08 | 2.93 | 2.83 | **1.83** | 2.98 |
| | cosd | 0.70 | **0.60** | 0.62 | 0.67 | 0.72 | 0.61 | 0.59 | **0.54** |
| Flower 64 × 64 | k-in | 0.97 | 1.39 | 1.72 | **2.00** | 1.08 | 1.51 | 1.48 | **2.03** |
| | cor | 2.26 | **2.41** | 2.31 | 2.24 | 2.27 | **2.43** | 2.24 | 2.26 |
| | cdl | 4.10 | 3.80 | **1.61** | 4.21 | 3.88 | 3.35 | **2.47** | 4.37 |
| | cor | 0.53 | 0.40 | **0.28** | 0.50 | 0.49 | **0.36** | **0.36** | 0.47 |
| ImageNet 224 × 224 | k-in | 0.73 | 1.05 | 1.38 | **1.52** | 0.72 | 1.02 | 1.45 | **1.50** |
| | cor | 1.73 | 1.90 | 2.08 | **2.15** | 1.73 | 1.89 | 2.10 | **2.14** |
| | cdl | 22.45 | 14.17 | 9.74 | **5.81** | 21.98 | 14.36 | 8.26 | **5.48** |
| | cosd | 0.93 | 0.76 | 0.49 | **0.41** | 0.93 | 0.76 | 0.47 | **0.41** |

Table 2: Evaluation of the top-k and manipulate attack on different dataset. We use $k = 20, 80$ and 1000 pixels, respectively for CIFAR-10, Flower, and ImageNet to ensure the ratio of $k$ over the total number of pixels (in the first column) is approximately consistent across the dataset. Each number in the table is computed by firstly taking the average scores over all evaluated images and then aggregating the results over different maximum allowed perturbation $\epsilon_\infty = 2, 4, 8, 16$ with the area under the metric curve. The bold font identifies the most robust method under each metric for each dataset.

| | | | TopK Attack | | | | Manipulate Attack | | | |
|---|---|---|---|---|---|---|---|---|---|---|
| | | | SM | IG | SG | UG | SM | IG | SG | UG |
| SSR (Ours) $\beta = 0.3$ | time | k-in | **0.68** | **1.05** | **1.04** | 2.95 | 0.80 | 1.18 | **1.26** | 2.95 |
| | 0.18h/e | cor | **2.21** | **2.37** | **2.15** | **2.26** | **2.24** | 2.40 | **2.21** | **2.29** |
| | acc. | cdl | **2.54** | 2.24 | **1.77** | **2.50** | 2.25 | **1.98** | **1.41** | **1.48** |
| | 81.2% | cosd | **0.45** | **0.35** | **0.35** | **0.39** | **0.41** | 0.33 | **0.31** | **0.38** |
| Mądry's [26] $\delta_2 = 0.25$ | time | k-in | 0.43 | 1.03 | 0.94 | **2.95** | **1.04** | **1.67** | 1.14 | **2.96** |
| | 0.24h/e | cor | 2.01 | 2.30 | 2.01 | 2.04 | 2.15 | **2.48** | 2.04 | 2.28 |
| | acc. | cdl | 3.09 | **2.20** | 1.84 | 3.08 | 4.76 | 3.26 | 1.75 | 3.59 |
| | 82.9% | cosd | 0.55 | 0.36 | 0.47 | 0.49 | 0.47 | **0.29** | 0.39 | 0.40 |
| IG-NORM [8] $\gamma = 0.1$ | time | k-in | 1.55 | 1.99 | 1.70 | 2.96 | 2.56 | 2.74 | 2.15 | 2.98 |
| | 0.44h/e | cor | 2.75 | 2.86 | 2.73 | 2.78 | 2.91 | 2.95 | 2.80 | 2.89 |
| | acc. | cdl | 1.25 | 0.91 | 1.18 | 1.22 | 1.51 | 1.18 | 0.96 | 1.48 |
| | 49.5% | cosd | 0.12 | 0.06 | 0.12 | 0.08 | 0.03 | 0.02 | 0.07 | 0.04 |

Table 3: Evaluation of the top-k and manipulate attack on CIFAR-10 with different training algorithm. The natural training is included in Table 2. We use $k = 20$. Each number in the table is computed by firstly taking the average scores over all evaluated images and then aggregating the results over different maximum allowed perturbation $\epsilon_\infty = 2, 4, 8, 16$ with the area under the metric curve shown in Fig. 2a. The bold font highlights the better one between Mądry's training and SSR. Per-epoch training time (time) and training accuracies (acc.) are listed on the second column.

### C1.1 Full Experiments

Full results of attribution attack for different attribution methods and training methods are show in Table 2 and 3. We include the preliminary results of variability experiments in Fig 6 for Saliency Map.Given the long-period of training and testing, full results are still under processing and will be released in the future. We include the preliminary results from other $\epsilon$ for adversarial training in Fig. 7. We use the data range in [0, 255] and $\epsilon$ is calculated under $\ell_2$ norms. We notice that when changing the data range to [0, 1], adversarial training can provide much better robustness in attribution attack as well. However, in this paper, given SSR is also trained on data range of [0, 255], we only compare the results on [0, 255] in this section. We are working on providing more comprehensive and detailed comparison in the future versions.

Figure 6: Variability experiments of SSR on Saliency Map (using the left y-axis). We trained 11 ResNet-20 models with $\beta = 0.3$ on CIFAR-10. The results of NAT, Madry and IG-NORM (using the right y-axis) are read from Table 2 and 3.

Figure 7: Evaluation of attribution attack on adversarially trained ResNet20 with $\ell_2$ balls.

## C2. Sensitivity of Hyper-parameters in SSR

**choice of scaling $s$.** We choose $s = 1e6$ from empirical tests for CIFAR-10 with data range in [0, 255]. We notice if data range [0, 1] is used, $s = 1$ is a good scaling factor. A proper $s$ can be found simply be setting $s = 1$ and $\beta = 1$ first to observe the scale of the regularization. The data range and dimensions of the input determines the choice of the scaling parameter $s$.

**choice of $\beta$.** We run a simple parameter seach for $\beta = 0.3, 0.5, 0.7, 0.9$. We train all the models on CIFAR-10 with 25 epochs and record the training accuracy in Tabel 4 and we run the *topk* attack on each model respectively on same 500 images where the results are shown in Fig. 8. Higher $\beta$ will require more training time to reach better performance and it may not necessarily produce better robustness than a small $\beta$ on some metrics, e.g. top-k intersection. Therefore, for the consideration of training time and robustness performance, we choose $\beta = 0.3$ in the Experiment II of Sec. 5.

| $\beta$ | 0.3 | 0.5 | 0.7 | 0.9 |
|---|---|---|---|---|
| train acc. | 0.81 | 0.78 | 0.70 | 0.62 |

Table 4: Training accuracies v.s. $\beta$ in SSR on CIFAR-10 ($s = 1e6$).

## C3. Extra Experiment of Attribution Attack with Adversarial Training

Adversarial training on ImageNet usually takes a long time with limited resources. Therefore, we only investigate how robust attribution maps are on a pre-trained ResNet-50[4] model and the results are shown in Table 5. It shows that if a user has enough time and GPU resources, using adversarial training can also produce considerably good robustness on gradient-based attributions.

## C4. Extra Experiments of Transfer Attack

**Setup of Experiments.** We describe the detailed setup for the Experiment III of Sec. 5. Experiments are conducted on 200 images from ImageNet dataset. A pre-trained[5] ResNet-50 model with standard training is used. We first generate perturbed images by attacking Saliency Map, then we evaluate

Figure 8: Attribution attack on ResNet-20 models with different hyper-parameter $\beta$ and the identical scaling term $s = 1e6$. The y-axis is the AUC score for each metric over $\epsilon_\infty = 2, 4, 8, 16$ as described in Fig 2a.

| | TopK | | | | Manipulate | | | |
|---|---|---|---|---|---|---|---|---|
| | SM | IG | SG | UG | SM | IG | SG | UG |
| k-in | 2.22 | 2.55 | 2.30 | 2.23 | 2.20 | 2.42 | 2.32 | 2.22 |
| cor | 2.69 | 2.87 | 2.72 | 2.71 | 2.69 | 2.82 | 2.73 | 2.70 |
| cdl | 5.70 | 2.89 | 5.55 | 5.75 | 5.69 | 3.27 | 5.23 | 5.69 |
| cosd | 0.31 | 0.18 | 0.22 | 0.21 | 0.32 | 0.22 | 0.22 | 0.22 |

Table 5: Robust model (adversarially trained with $\epsilon_\infty = 8$) evaluation of the top-k, manipulate and mass center attack on ImageNet dataset. We use $k = 1000$ pixels. Each number in the table is computed by firstly taking the average scores over all evaluated images and then aggregating the results over different maximum allowed perturbation $\epsilon_\infty = 2, 4, 8, 16$ with the area under the metric curve. The bold font identifies the most robust method under each metric.

the difference between all attribution maps on the original and perturbed images. The number of iterations in the original saliency map attack is 50, number of steps for IG,SG and UG is 50. Top-K intersection (K=1000), correlation, mass center dislocation and cosine distance are used to measure the difference. Each number in the table is computed by firstly taking the average scores over all evaluated images and then aggregating the results over different maximum allowed perturbation $\epsilon_\infty = 2, 4, 8, 16$ with the area under the metric curve.

In the Experiment III of Sec. 5, we show the transferability of attribution attack. We here further show the transfer attack with *mass center* attack of Saliency Map on all other attribution methods in Table 6. Compared with Smooth Gradient and Uniform Gradient, Integrated Gradient also has larger dissimilarity on all metrics.

|       | Mass center |       |       |       |
|-------|-------------|-------|-------|-------|
|       | SM          | IG    | SG    | UG    |
| k-in  | 0.71        | 1.01  | 1.44  | 1.54  |
| cor   | 1.73        | 1.87  | 2.12  | 2.17  |
| cdl   | 24.41       | 17.34 | 8.99  | 5.09  |
| cosd  | 1.01        | 0.79  | 0.47  | 0.41  |

Table 6: Transferability evaluation of mass center attack. We attack on the Saliency Map (SM) and evaluate the difference between all attribution maps on the original and perturbed images.

Figure 9: Visualization results of 7 baseline methods.

# Supplementary Material D

## D1. Visual Comparison

In this section, besides the robustness, we compare the visualizaiton of Uniform Gradient with several existing methods shown in Fig 9. The visualization shows that Uniform Gradient is also able to visually denoise the original Saliency Map.

| Input | Radius=2 | Radius=4 | Radius=8 | Radius=16 | Radius=32 | Radius=64 |
|-------|----------|----------|----------|-----------|-----------|-----------|

Figure 10: Visualization results of Uniform Gradient with different noise radius.

## D2: UniGrad under different smoothing radius

Choosing the noise radius $r$ is also a hyper-parameter tuning process. We provide visualization of the same input with different noise radius from 2 to 64 under 0-255 scale with 50 times sampling to approximate the expectation in Fig 10. When the noise radius is too low, it can not denoise the Saliency Map while if the noise radius is too high, the attribution map becomes "too dark".

## Footnotes

[1] we use the implementation on https://docs.scipy.org

[2] code is availabe on https://github.com/amiratag/InterpretationFragility

[3] https://github.com/jfc43/robust-attribution-regularization

[4]we use the released weight file from https://github.com/MadryLab/robustness

[5]we use the released weight file from https://github.com/MadryLab/robustness