[Reviews · NeurIPS 2020]

Review 1

Summary and Contributions: This paper analyzes the idea of attribution robustness and introduces two methods -- one for pretrained models, one for learning new models -- for increasing it. The method for pretrained models is based on integrating saliency maps over a uniform distribution (rather than a Gaussian) centered at the input. The method for learning new models is based on minimizing the largest eigenvalue of (an approximation of) the input Hessian (which reduces an upper bound on the worst-case attribution dissimilarity).

Strengths: The theoretical grounding of the submission seems sound, as does the general strategy of the empirical evaluation. I think the contribution is significant and relevant to the NeurIPS community.

Weaknesses: I have one main reservation about the empirical evaluation, which is that for the adversarial training and IG-NORM baselines, results are only shown for a single hyperparameter setting -- even though increasing the hyperparameters means increasing robustness (at the cost of some test set accuracy). Although I understand this is a perennial challenge for adversarial robustness method comparison, it seems like a fairer comparison would do hyperparameter sweeps for _all_ methods, plotting test set accuracy vs. robustness, and seeing whether any method consistently has higher robustness for the same test set accuracy. I also have a few concerns about novelty, which I'll list below.

Correctness: The submission's analysis of the problem of attributional robustness seems correct, as do the methods it introduces. The empirical methodology seems mostly correct except (arguably) for the issue I discuss above.

Clarity: The paper is clearly written and easy to follow. The example in Figure 1 is especially nice.

Relation to Prior Work: Overall its discussion of related work is clear, but I am concerned that https://arxiv.org/abs/1901.09392 from NeurIPS 2019 is extremely similar. That work has the same two core contributions: it introduces new methods of locally integrating saliency maps to maximize the robustness of pretrained model attributions, and it introduces an approximate Hessian regularization objective for training new models. Although both of these components have slightly different formulations, I think it would be good to clarify in the submission (and the rebuttal) how this work differs. Also, the authors cite Singh et al. (reference 39, https://arxiv.org/abs/1911.13073) as a different kind of method, but it seems like it might actually be another baseline they could directly compare against (though this could be only true as of a recent revision, judging by the arXiv version history).

Reproducibility: Yes

Additional Feedback: Although I think this paper makes valuable contributions, I'm on the fence, and my initial vote is going to be slightly against. I'm very much open to increasing my score based on the rebuttal and ensuing reviewer discussion though, as I could be wrong, and there are also many things it seems like the authors could do to improve the submission within the time-frame of camera-ready preparation. Of these, the most impactful seem like including additional experiments that compare robustness vs. accuracy for many settings of hyperparameters (for the baselines as well as the main method), and clarifying the differences with https://arxiv.org/abs/1901.09392 (maybe including it as an additional baseline, for both the pretrained and newly trained contributions). === Update in response to author feedback: I'm satisfied with the author response, and feel that the additional experiments and discussion of related work brought up by all the reviewers will make it much stronger. Therefore, I'm increasing my score accordingly.


Review 2

Summary and Contributions: The paper provides a theoretical analysis for the problem of interpretation fragility, a well-known problem with gradient-based interpretability methods. It is proved that given a large enough noise parameter for smooth-grad, the Lipschitz constant for the interpretability function can become arbitrarily small, meaning that, an l2 norm-bounded perturbation will not be able to change the interpretation drastically. ====================== Thanks for your rebuttal discussion. I still think the addressed problem is important, however, there can be more work done. I will not change my score.

Strengths: The theoretical insight of why smoothgrad saliency maps seem to be more robust is a useful and necessary addition to https://arxiv.org/abs/1906.07983 for a better understanding of the interpretation fragility phenomenon. As a result, they are able to train a smoothness-induced model in order to have more robust interpretatins. - The discussion is general and not limited to a specific data modality.

Weaknesses: The main weakness with the paper is the that whether the contributions are enough for a NeurIPS paper or not. - The proposed SSR algorithm seems way weaker than IG-Norm while not being much (order of magnitude) faster. - The relationship between interpretability and 2nd order smoothness has been already (not deeply) discussed in the appendix J of https://arxiv.org/pdf/1905.12105.pdf and the paper does not mention its differential contribution. - Experiment I results have already been shown in previous literature.

Correctness: The theoretical discussions and the emprirical results are sound.

Clarity: The writing is very clear and easy to follow. Typo: Line 254 an recent proposed

Relation to Prior Work: Relationship to the previous work is mostly discussed clearly. Not enough credit is given to how much the work is related to https://arxiv.org/pdf/1905.12105.pdf IMHO.

Reproducibility: Yes

Additional Feedback: The idea seems like a 2nd-order extension of https://arxiv.org/abs/1412.6572 method for prediction robustness. It would be interesting to be discussed.


Review 3

Summary and Contributions: This paper theoretically analyzed the robustness of some feature attribution methods, and based on this, proposed a technique for robustness against feature attribution attacks. The transferability of local perturbation was discussed, and it was shown that the proposed method was efficient through the regularization of the input gradients. The authors also proposed an alternative stochastic smoothing method dubbed as Uniform Gradient, and showed that this technique improved attribution robustness.

Strengths: + The geometrical understanding of the robustness of attribution methods is theoretically verified with Lipschitz continuity. + Based on this understanding, they analyzed the possibility of to transfer attack between Saliency Map and Smooth Gradient. + The authors have effectively implemented a method for obtaining robust attributions based on their theoretical analysis, including retraining (SSR) or post-hoc analysis (UG).

Weaknesses: Major: - This paper covered theoretical robustness and transferability only for Saliency Map (simple gradient) and Smooth Gradient. It would be nice to deal with other attribution methods (e.g. Integrated Gradient) mentioned in the paper. - Singla et al’s closed-form formula was used when regularizing the spectral norm of the input Hessian to implement SSR. One might wonder why the authors used this method to approximate the spectral norm, not the method of [1] or [2], so it seems to need justification and analysis. [1] Seyed-Mohsen Moosavi-Dezfooli et al. “Robustness via curvature regularization, and vice versa.” IEEE/CVF Conference on Computer Vision and Pattern Recognition (CVPR). 2019. [2] Yuichi Yoshida et al. “Spectral Norm Regularization for Improving the Generalizability of Deep Learning.” arXiv. 2019. - Please provide "generality" experiments to black-box attacks. Define two different models: model A and model B. Generate attack samples using model A, and evaluate on model B, trained under SSR, [27], or [10]. A minor concern is as follows: - The manuscript has many typos. In addition, inconsistent notations in proofs should be corrected.

Correctness: Yes

Clarity: In Theorem 2, it seems to require a detailed explanation of the transition from local Lipschitzness to global robustness. I think the reason, as I understood it, is that the dependency on the boundary of x’ disappears in the proofs.

Relation to Prior Work: I think the theoretical analysis of attribution robustness has proper contributions over the previous methods.

Reproducibility: Yes

Additional Feedback: 1. Tables 1 and 2 seem too complicated. Visualization using something like a bar graph seems to help people understand. 2. It would also be interesting to consider robustness for model-level manipulation attack (e.g., [17]) rather than image-level manipulation. ======= Post author feedback ======= Thanks a lot for the authors' reply. I have read all the comments from other reviewers and the author feedback. I would keep my original rating.


Review 4

Summary and Contributions: The authors provide a theoretical analysis of the relationship between smoothness of the model’s gradient and the model’s robustness to attribution attacks as well as the ability of the attack to transfer across different attribution methods. In addition to that, by relying on these theoretical results, the authors provide practical approaches for making attribution models more robust. Lastly, they test these solutions in a number of experiments, demonstrating their promise.

Strengths: The paper addresses a highly relevant topic, with potential implications for a large part of the ML community. General research plan is sound. The authors identify a problem, provide theoretical analysis of its roots, and provide two empirical methods, both justified theoretically, and tested in a number of experiments. Another nice practical feature of the proposed methods is that one of them is post-hoc, while another requires an intervention during training. Thus, an ML practitioner can choose between them depending on the resources available and the stage of the development process. The approach is sufficiently novel. The structure of the paper is largely that of generalization and adaptation of known results to a new domain, which may limit the perceived novelty. I believe, however, the originality, while not being the strongest part of this paper, is certainly sufficient for this type of contribution and is up to the standards of NeurIPS conference.

Weaknesses: There are a number of issues that negatively affect the overall quality of the contribution. There are imperfections in every element of assessment (clarity, correctness, relation to prior work), but only the correctness issues are relatively serious and not easily fixed. I detail these issues in the respective sections of my review.

Correctness: Overall, the theoretical part of the contribution is sound. The general research plan allows to evaluate the promise of the method. The three experiments cover sufficiently many angles of model performance analysis. Unfortunately, the execution and presentation of these experiments is not perfect. Most importantly, there are no variability measures in the provided results. It severely complicates the interpretation of the obtained results and makes it difficult to objectively assess the promise of the method. If re-running experiments many times is computationally prohibitive, I believe that it is necessary to include, at least, some bootstrap-based estimates (bootstrapping over images). Experiment 1 is especially difficult to interpret as the proposed methods do not uniformly outperform the alternatives. It appears, however, that the results are very promising. What looks especially promising is that it seems that the proposed methods are not outperforming alternatives mostly in cases where all methods perform more or less the same. It could even be random error, but it’s, again, hard to judge because of the absence of variability estimates. It may also be very helpful to report more summary statistics (such as, for example, the average improvement across datasets). Overall, I believe that more work should be done in terms of discussing and interpreting these results. These considerations would have been, in my opinion, disqualifying if the paper was purely empirical. The presence of a fairly thorough theoretical analysis and the general consistency of the results with that analysis make me believe that the paper remains above acceptance threshold, even though the imperfections of the experimental setup place it into a “borderline” category.

Clarity: In terms of clarity, it seems that the paper is uneven. The first sections look very polished and are a pleasure to read. At the same time, the quality of writing seems to deteriorate in the second half of the paper, which is marked by an increased number of typos and decreased readability (please see a list of typos in the suggestions section).

Relation to Prior Work: The discussion of the relationship to the prior work appears to be overly short and merely lists a number of proposed methods, without analyzing their benefits and limitations. This is largely compensated by the fact that much of related work is mentioned or discussed throughout the text. It seems, however, that re-structuring the “related work” section may be beneficial.

Reproducibility: Yes

Additional Feedback: -The table layout in Table 2 is very confusing. It may be better to re-organize the second column (the one listing time and accuracy). In the very least, it’s important to remove row separation in it, since it serves no semantic purpose. -I personally found the illustration on Figure 1 neither necessary nor very illuminating. I.e. the general idea was already clear, thanks to a detailed verbal description, while the Figure 1 not only did not add to this understanding, but was rather confusing. I don’t think it’s necessary to remove it, but it may be beneficial, since it would free up some space for potentially more important sections (e.g. experiment 1 discussion or expanding “related work” section). -I am not sure if that would be easy to arrange, but it may be possible to make figure 2b easier to interpret “at a glance”, if the attack objective was set to some arbitrary mask (e.g. a picture of a large plus sign), instead of achieving maximal distance from the original attribution. To clarify: the figure is fine as is, and I don’t want the authors to feel any pressure to do it. This is just a suggestion that may improve experience for future readers. Typos: 188 - locally robustness -> locally robust 249 - reply -> rely?, perceptional -> easier to perceive, human -> humans 254 - an -> a, recent -> recently. In general, that sentence appears confusingly phrased 261 - discovery -> discover 262 - slow -> slower ## I have read the author's response. I am happy to hear that some of my comments were useful. Unfortunately (since the rebuttal period is short and since the rebuttal is limited to one page), there is no way for me to see how successfully the authors address the issues that I and other reviewers mentioned (for example, even though the authors plan to include the variability measures into the final version, the results are not provided in the rebuttal). I, therefore, intend to keep my score unchanged.

[Author Response · NeurIPS 2020]

**All Reviewers** We appreciate the helpful feedback of our reviewers and the pointers to relevant recent work. We agree that some of our tables and layout could be better designed and are grateful for the suggestions for doing so. We apologize for the typos identified and will make sure that the paper is suitably polished in the final version.

**Reviewer #1** **Parameter Sweeping.** We agree that a parameter sweep in the adversarial training experiments will produce fairer results. When choosing $\epsilon$, we used a common setting ($\ell_2, \epsilon = 0.25$) for data scaled to $[0, 1]$, and in the final version, can instead sweep in increments of $0.1$. Experiment II is computationally intensive so we are unable produce these results before the rebuttal deadline but we expect that increasing $\epsilon$ or maximum attack steps in each iteration may increase the robustness but also increase the training budget for adversarial training. We are working on additional experiments with different hyper-parameters to build scatter plots of the robustness against accuracy. **Novelty Concerns**. 1) We appreciate that the reviewer brings Yeh et al (2019) to our attention. We find one major difference is that Yeh et al (2019) is motivated by evaluating attribution methods (i.e., no attacks are evaluated in their work), whereas our focus is on robustness to attacks. Nonetheless, in theory SENS-MAX is indeed another way to incorporate our local robustness $\lambda$. However, using Monte Carlo to approximate SENS-MAX is different from performing gradient descent to attack attributions, and especially in high-dimensional image spaces, an adversarial example may not be easily sampled. Another difference is that while Yeh et al (2019) proposes Hessian regularization as a possible remedy, we implement and evaluate its effectiveness in the adversarial setting. We will update the paper to reflect this discussion. 2) We find the 3rd version of Singh et al (2020) is the one we cited. There are significant differences between the newest version, which was not available the time of submission, and the one we cited. We will update the paper to discuss the latest version, as well as adding the relevant comparisons to our experiments.

**Reviewer #2** **Comparison with IG-NORM** We agree that IG-NORM tends to show greater robustness, but this comes at an appreciable cost in accuracy. In preparing our experiments, we also found that it is quite sensitive to initialization, leading to variability in model fitness for a given allocation of training time in epochs (mentioned on lines 261-263). These tradeoffs may be significant for practical settings. **Appendix J of Levine et al (2019).** We appreciate bringing this appendix to our attention. We believe our work can be viewed as a further exploration of the results shown in in Figure 13, giving geometric intuitions along with new theory and experiments that might shed fresh light on those results. **Novelty of Experiment I.** Although we are curious about which references the reviewer has in mind, to the best of our knowledge prior work has not shown the effectiveness of stochastic smoothing on the Dombrowski et al (2019) attack, and we are not aware of references that report measurements of a technique similar to our Uniform Gradient. If we are mistaken, we would appreciate pointers to the relevant work.

**Reviewer #3** **Analysis for IG.** We agree that including more analysis of IG is an interesting direction, with relevance to practice given the popularity of that method. However, characterizing IG under the same assumptions that we made for SG requires considering the global geometry of the model between an arbitrary baseline and the input. This is part of our ongoing work, but is a significant addition that would be difficult to present adequately in a single short paper. **Other Hessian Approximations.** 1) Thank you for making us aware of Moosavi-Dezfooli et al (2018). If we had known of it at the time of submission, we might still have used Singla et al (2019), as the two-step approximation used by Moosavi-Dezfooli et al (2018), along with the fact that we do not require approximation over a Gaussian but instead a single point, suggests that Singla et al (2019) is a better fit in terms of efficiency for our needs. 2) We expect that Singla et al (2019) provides a closer approximation than Yoshida et al (2017), which regularizes the aggregate spectral norms of weights at each layer, giving a loose upper-bound of the input Hessian spectrum. However, we will include discussion of these methods in the paper, as well as appendices containing experimental comparisons. **Generality Experiments.** We suspect that a black-box attribution adversary may not be as meaningful in practice as a black-box label adversary, except when it is reasonable to assume that models trained on the same distribution are expected to have (approximately) identical attributions for given test points. However, we agree that these experiments may generate interesting results, and will report on them in an appendix in future versions of the paper.

**Reviewer #4** **Variability Measurement.** We appreciate the discussion on variability tests and agree that they will strengthen our conclusions. We note that some of the attacks we measured (e.g., SM and IG) are deterministic and we have not observed significant variability in those parts of our empirical analysis when hyper-parameters are fixed. As mentioned in the review, our experiments are computationally intensive. Therefore we were unable to include multiple trials at the time of submission; we will report results over multiple trials in the final version along with a discussion of variability. **Results and Prior Work.** In the experiment section, we agree that more analysis and interpretations about our empirical results can make the conclusions more convincing. Also, as mentioned by previous reviewers, we notice that there are more interesting work of which we are not aware at the moment of submission. We will re-structure the "Related Work" section based on feedback from all reviewers. **Writing and Layout.** We appreciate your (very) helpful suggestions about reorganizing some of our figures and tables; we agree that Table 2 can be confusing, and we plan to re-design Tables 1 and 2 based on your feedback.

[Meta-Review · NeurIPS 2020]

The paper found that a way to improve the robustness of gradient-based attribution maps is by smoothing the gradient attribution maps. They also propose a training regularizer (i.e. minimizing the largest eigenvalue of the Hessian w.r.t. input). All reviewers found the theory is sound and consistent with empirical results. This is a good paper and I recommend accept. I'd suggest the authors to also discuss the connection (and differentiate) between the robustness of attribution maps vs. the robustness of classifiers (e.g. would the proposed regularizer improve model robustness?).